# A Fungicide, Fludioxonil, Formed the Polyploid Giant Cancer Cells and Induced Metastasis and Stemness in MDA-MB-231 Triple-Negative Breast Cancer Cells

**DOI:** 10.3390/ijms25169024

**Published:** 2024-08-20

**Authors:** Ryeo-Eun Go, Su-Min Seong, Youngdong Choi, Kyung-Chul Choi

**Affiliations:** Laboratory of Biochemistry and Immunology, College of Veterinary Medicine, Chungbuk National University, Cheongju 28644, Chungbuk, Republic of Korea; gmyich@naver.com (R.-E.G.); sumin98422@gmail.com (S.-M.S.); youngdong2469@gmail.com (Y.C.)

**Keywords:** fludioxonil, fungicide, TNBC, polyploid giant cancer cell

## Abstract

Fludioxonil, an antifungal agent used as a pesticide, leaves a measurable residue in fruits and vegetables. It has been identified to cause endocrine disruption, interrupt normal development, and cause various diseases such as cancers. In this study, fludioxonil was examined for its effects on the development and metastasis of breast cancer cells. On fludioxonil exposure (10^−5^ M) for 72 h, mutant p53 (mutp53) MDA-MB-231 triple-negative breast cancer (TNBC) cells significantly inhibited cell viability and developed into polyploid giant cancer cells (PGCCs), with an increase in the number of nuclei and expansion in the cell body size. Fludioxonil exposure disrupted the normal cell cycle phase ratio, resulting in a new peak. In addition, PGCCs showed greater motility than the control and were resistant to anticancer drugs, i.e., doxorubicin, cisplatin, and 5-fluorouracil. Cyclin E1, nuclear factor kappa B (NF-κB), and p53 expressions were remarkably increased, and the expression of cell cycle-, epithelial–mesenchymal-transition (EMT)-, and cancer stemness-related proteins were increased in the PGCCs. The daughter cells obtained from PGCCs had the single nucleus but maintained their enlarged cell size and showed greater cell migration ability and resistance to the anticancer agents. Consequently, fludioxonil accumulated Cyclin E1 and promoted the inflammatory cytokine-enriched microenvironment through the up-regulation of TNF and NF-κB which led to the transformation to PGCCs via abnormal cell cycles such as mitotic delay and mitotic slippage in mutp53 TNBC MDA-MB-231 cells. PGCCs and their daughter cells exhibited significant migration ability, chemo-resistance, and cancer stemness. These results strongly suggest that fludioxonil, as an inducer of potential genotoxicity, may induce the formation of PGCCs, leading to the formation of metastatic and stem cell-like breast cancer cells.

## 1. Introduction

Polyploidy is a condition in which the cells of an organism have more than one pair of homologous chromosomes. Polyploidy is common in plants, fungi, and some animal cells. Polyploidy is a classic mechanism of biological evolution and plays a critical role in the normal growth, development, and differentiation of macrophages, hepatocytes, osteoclasts, cardiomyocytes, and muscle cells, and in response to adverse stimulation [1,2,3]. However, abnormal polyploidy commonly appears in pathophysiological states such as in chronic hepatitis, several age-related diseases, and cancer [3,4].

Specifically, polyploidy in cancer manifests as polyploid giant cancer cells (PGCCs) that promote heterogeneity, tumorigenesis, and resistance to therapy. PGCCs are known to develop from a diploid tumor cell in response to treatment or microenvironment stress, including chemotherapy, radiotherapy, aurora kinase inhibitor therapy, hypoxia, and chemical exposure [1,5,6]. Polyploidization occurs due to abnormal cell division, either during mitosis or from the failure of chromosomes to separate during meiosis. It can occur via endocycling, endomitosis, cytokinetic failure, or cell–cell fusion resulting in PGCCs, which are known to produce daughter cells by budding or bursting because of abnormal mitotic division. These daughter cells have been studied in the context of their cancerous characteristics such as invasion, migration, and metastasis into osteoclast-like cells, cancer stromal cells, cancer-associated fibroblast, and malignant-cancer stem cells (CSCs) [7]. They are associated with tumor initiation and progression, therapy resistance, and metastasis in various carcinomas, including breast, ovarian, prostate, and lung carcinomas [8,9].

Metastasis is the migration of cells from their first site to a different body location and the generation of new tumor colonies. To initiate metastasis, cancer cells need to possess a set of characteristics including epithelial–mesenchymal transition (EMT), migration and survival abilities, and proliferation despite a limited environment [10,11]. The metastasis of tumor cells starts with EMT when tumor cells are transformed into sharp shapes to infiltrate the circulatory and lymphatic systems. With a migratory ability, these cells spread from the initial tumor site, infiltrate through basement membranes and endothelial walls, and settle in other organs for survival and proliferation during the genesis and progression of cancer. Subsequently, the proliferated cells sens a signal to create blood vessels and thus provide nourishment for cancer development [12,13,14]. Generally, the level of cell-cell adhesion protein, N-cadherin, and filamentous cytoskeletal protein, Vimentin are increased, while the levels of cell–cell tight junction proteins, E-cadherin and Occludin are decreased to mediate EMT-induced tumor invasion [14].

Furthermore, CSCs appear to play a fundamental role in the pathogenesis of tumor recurrence, distant metastasis, and therapy resistance. According to studies, tumors comprise functionally heterogeneous subgroups of cells and the cells with stemness characteristics are a rare sub-population. They have stem cell-like properties, play a crucial role in EMT, and are capable of self-renewal, tumor initiation, high proliferation, and/or are chemotherapy resistant. They also show additional characteristic properties, such as the expression of cell surface markers (CD44 and CD133), high activity of the enzyme aldehyde dehydrogenase (ALDH), and/or the dysregulation of the self-renewal signaling pathway (Wnt, Notch, or Hedgehog). In addition, the biological activities of CSCs are regulated by several pluripotent transcriptional factors, such as the sex-determining region Y-box 2 (SOX2), homobox protein NANOG, octamer-binding transcription factor 4 (OCT4), Krüppel-like factor 4 (KLF4), and myelocytomatosis oncogene (MYC) [15,16,17,18].

Fludioxonil (4-(2,2-difluoro-1,3-benzodioxol-4-yl)-1H-pyrrole-3-carbonitrile) is an antifungal agent used worldwide to control gray mold in a variety of agricultural applications. In an earlier study by the authors, the effects of fludioxonil in MCF-7 CV breast cancer cells were studied in cellular and xenografted mouse models. In the cellular model, the cell viability and migration induced by fludioxonil was similar to that in the estradiol (E2)-positive group through the upregulation of the cell cycle- and EMT-related proteins. In the xenografted mouse model, the fludioxonil exposure resulted in an increase in the tumor size, which was up to 40% of the E2 tumor size. In addition, the expressions of the EMT marker N-cadherin and metastasis marker Cathepsin D in the tumor were significantly increased by the fludioxonil exposure. In this study, the effects of fludioxonil were more clearly observed in the xenografted mouse model than in the cellular model. In other words, these results could be interpreted to state that fludioxonil induced tumor growth by the E2-dependent and -independent pathways [19]. In other studies, fludioxonil was observed to potentially act as an endocrine disruptor (estrogen receptor α [ERα], aryl-hydrocarbon receptor [AhR], and weak androgen receptor [AR] agonist) on hormone-dependent cancer [20]. Moreover, fludioxonil has also been suggested to work in other ways as an endocrine disruptor on hormone-dependent cancer [21]. In the animal model, fludioxonil suppressed the synthesis of prostaglandin D2, which plays an essential role in expressing male sexual differentiation in the fetus [22]. These results indicate fludioxonil’s negative effects on organisms as an endocrine disruptor.

Furthermore, other adverse effects of fludioxonil on organisms have also been published. Fludioxonil disrupted the development of the normal nervous system and induced cytoskeleton disruption, DNA damage, and apoptosis via oxidative stress in rat glioma cells [23]. Additionally, fludioxonil inhibited the viability of immune cells (T and B lymphocytes) and induced cell apoptosis [24]. Catalase is an enzyme with antioxidant properties. In a study on bovine liver catalase activity, fludioxonil has been shown to inhibit this enzyme non-competitively [25]. In addition, exposure to fludioxonil inhibited the cell viability of mouse embryonic stem cells, the formation of mouse embryoid bodies, and the development of cardiomyocyte differentiation in the early stage [26]. Among the genotoxic mixture of five compounds present in the diet of the French population, fludioxonil was found to be genotoxic for HepG2 cells alone at a concentration of 20 μM [27]. These results significantly increase the suspicion regarding the safety of the fungicide fludioxonil.

An unusual finding of our earlier study was that the morphology of the MDA-MB-231 breast cancer changes after exposure to fludioxonil. It was also noted that this change is related to PGCCs through diverse previous studies. Therefore, the current study aimed to examine the effects of fludioxonil, a commonly used pesticide, in inducing metastatic cancer by forming PGCCs in triple-negative breast cancer (TNBC) MDA-MB-231 cells.

## 2. Results

### 2.1. Effects of Fludioxonil on the Cell Viability of Breast Cancer Cell Lines

To compare the effects of fludioxonil on the viability of breast cancer cells, the water-soluble tetrazolium (WST) cell viability assay was performed after exposing the cells to E2 (10^−9^ M) or fludioxonil (10^−5^ and 10^−6^ M) for 72 h. In T47D and MCF-7 breast cancer cell lines expressing ERα, cell viability was significantly higher after treatment with E2 and fludioxonil (10^−5^ M) compared with the control (Figure 1B,C). However, there was no change in cell viability of the MDA-MB-231 cells after treatment with E2 and it was significantly lower after treatment with fludioxonil (10^−5^ M) for 72 h compared to the control (Figure 1A). MDA-MB-231 cells treated with fludioxonil (10^−5^ M) showed relatively lower cell viability than the control after exposure for 96 h (Figure 1E). After treatment with fludioxonil (10^−5^ M) for 72 h, the MCF-7 cells showed increased cell density without a change in cell morphology. However, the cell morphology of the MDA-MB-231 cells was identified as an expanded cell body, as compared to the control (Figure 1D).

To confirm the changes to the cell morphology in detail, the cells were stained using ActinGreen and the Hoechst dye (Figure 2A). The cell morphology of the MDA-MB-231 cells was changed to a hetero-, enlarged, and multinucleated shape. The cell number after fludioxonil (10^−5^ M) treatment for 72 h did not significantly increase compared to the control (Figure 2B). In contrast, the average size of the nuclei (Figure 2C) and the cell body size (Figure 2D) increased significantly, 1.12 and 1.33 times, respectively, compared with the control. These findings indicate that the viability of the cells after treatment with fludioxonil was similar to that with E2 in ER-expressing breast cell lines, but fludioxonil induced morphological changes without increasing cell viability in the MDA-MB-231 breast cancer cells.

### 2.2. Transformation to Polyploid Giant Cancer Cells

To confirm the transformation of the cancer cells to PGCCs, the cells were monitored using the live-cell image system for 84 h. After treatment with fludioxonil (10^−5^ M) for 72 h, a cell cycle analysis was conducted using fluorescence-activated cell sorting (FACS) to confirm the percentage of PGCCs compared to the control. The morphology and cell division in real time were shot using the live-cell image system (Appendix A). In the control, one cell normally divided into two daughter cells (Figure 3A, left panel, blue arrow, and line). If one cell temporarily had two nuclei because it could not divide, it was confirmed that the cell divided into four daughter cells in the next cell cycle (Figure 3A, left panel, red arrow, and line). Finally, each cell had only one nucleus. The average round-up time for the division into daughter cells was 2.4~2.7 h in the control (Figure 3B). On the other hand, the single cells treated with fludioxonil (10^−5^ M) did not divide into two cells after mitosis and had two or more nuclei in a cell (Figure 3A, right panel, yellow arrow, and line). Also, the average round-up time for daughter cell division was 9.4~12.7 h, which was significantly longer than the control (Figure 3B). To visually indicate the position of the nucleus, the nuclei were dyed using a color or fluorescence tracer and a real-time video of the living cells was recorded, but the change due to fludioxonil did not appear after staining the cell nuclei.

The cell cycle phase of the control consisted of the G0/G1 phase, the S-phase, and the G2/M phase, which were approximately in the average ratio of 42%, 28%, and 30%, respectively (Figure 3C,D). In contrast, the cell cycle phase of the fludioxonil (10^−5^ M)-treated group added a new peak (green) next to the G2/M phase, which means a multi-ploid cell (Figure 3C). Ultimately, the cell cycle phase at 24 h of the fludioxonil (10^−5^ M)-treated group consisted of the G0/G1 phase, S-phase, G2/M phase, and the multi-ploid phase, which were approximately in the ratio of 32%, 12%, 28%, and 28%, respectively. After fludioxonil treatment for 72 h, the ratio of the G0/G1 phase (21%) and the S-phase (6%) significantly decreased, and the ratio of the aneuploidy phase (41%) significantly increased (Figure 3E). At 96 h, the ratio of the cell-cycle peak was not significantly changed, but the cell morphology showed a more enlarged body, increased number of nuclei in a cell, and induction of blebbing-like lipid droplets around the nucleus than at 72 h.

The transformation to PGCCs induced a change in the expression of the cell cycle-related proteins (Figure 3F,G). The expression of G2/M phase-related proteins was increased slightly but significantly, with Cyclin A1 being on average 1.2 times higher and Cyclin B1 1.4 times higher than the control. Specifically, the expression of Cyclin E1 (average 1.9-fold higher), associated with the S-phase (chromosome replication), was significantly increased compared with the control. At the same time, the expression of Cyclin D1, the G0/Gl phase stimulator, did not change. The expression of p53 accumulated and increased 2.3 times in the PGCCs induced by fludioxonil exposure. These results supported the premise that the exposure of MDA-MB-231 breast cancer cells to fludioxonil (10^−5^ M) transformed the cells into PGCCs by inducing their development into multi-ploid cells, as a result of the increased expression of cell cycle-related proteins.

### 2.3. Cell Migration and Resistance to Anticancer Drugs in PGCCs Formed by Exposure to Fludioxonil

PGCCs-Flu were produced by exposure to fludioxonil (10^−5^ M) for 72 h. In addition, the migratory ability of PGCCs-Flu and resistance to anticancer drugs were evaluated by performing a cell scratch and cell viability assay after seeding PGCCs-Flu. Cell proliferation in the scratch assay was inhibited by treatment with mitomycin C to prevent false results due to the high cell density. In addition, because the cell body area of the PGCCs-Flu was larger by about 1.33 times compared to the control, the number of cells in each well of the six-well plate was adjusted.

The average pixel percentage of the scratch area at 48 h was 31% (control) and 18% (PGCCs-Flu). The scratch area of the PGCCs-Flu quickly decreased when compared with the control (Figure 4A,B). PGCCs-Flu were insensitive to the apoptosis or necrosis of cells by treatment of doxorubicin (Figure 4C), 5-FU (Figure 4D), or cisplatin (Figure 4E) compared with the control, indicating the chemo-resistance of anticancer drugs. Interestingly, chemo-resistance of PGCCs-Flu were highly induced at higher concentrations of anticancer drugs than the control. In the control, after the doxorubicin treatment (at half maximal inhibitory concentration, IC50 of 2.8 μM), the cell viability drastically decreased. In the PGCCs-Flu at IC50 of 10.7 μM of doxorubicin, the cell viability was higher than the control. IC_50_ for 5-fluorouracil (5-FU) in the control and PGCCs-Flu were 39.3 μM and 73.6 μM, respectively, while the corresponding IC_50_ for cisplatin were 27.1 μM and 50.0 μM, respectively. These results indicate the high resistance of PGCCs-Flu to anticancer agents. The variance (about 3.8-fold) of the doxorubicin IC_50_ value with the control and PGCCs-Flu was highest among the three agents, suggesting that more sensitive drugs in cells were developed at higher resistance in PGCCs-Flu.

The transformation to PGCCs changed the expression of the EMT- and CSC-related proteins (Figure 4F,G). The expression of E-cadherin, the epithelial cell marker of EMT, was slightly decreased, and the expression of Vimentin, the mesenchymal marker, was significantly increased in the PGCCs. The expressions of NANOG and SOX as an oncogene to promote carcinogenesis were significantly increased in PGCCs. These results indicated that the transformation of the cells to PGCCs stimulated the migration of the cells and resulted in resistance to anticancer drugs through the regulation of the EMT- and CSC-related proteins.

### 2.4. Effects of Fludioxonil on Change to Cell Pathways

Next-generation sequencing (NGS) was performed to confirm the genetic changes in the mechanisms associated with the formation of PGCCs due to fludioxonil exposure. Gene Ontology (GO pathway) and Kyoto Encyclopedia of Genes and Genomes (KEGG) pathway analyses were performed using DEGs obtained from the RNA sequencing raw data of the NGS.

The DEGs involved in the ‘biological process’ were further analyzed using the GO pathways, and 10 pathways were representatively predicted (Figure 5A). The highest category was the ‘regulation of cell population proliferation’, followed by the pathways meant for the process of diverse cell activity by a biotic stimulus such as movement, secretion, enzyme production, and gene expression. Using the KEGG pathway, ‘cytokine–cytokine receptor interaction’ was the highest category represented in 10 pathways (Figure 5C). The other pathways were related to inflammation and the microenvironment of cancer cells. The Tree diagram has been summarized as a hierarchical clustering tree of the correlation among significant pathways listed in the Enrichment tab. The pathways with many shared genes were clustered closely, and the bigger dots indicate more significant *p*-values. In the Tree diagram of the GO (Figure 5B) and KEGG (Figure 5D) pathways, the cytokine-related pathways had the highest correlation in other pathways.

Based on the KEGG pathway results and the raw data of the DEGs, the expression of some genes was selected and confirmed using quantitative polymerase chain reaction (qPCR). The expression level of the tumor necrosis factor (TNF) signaling-related genes showed a trend similar to the log2 FC of the DEGs, although the absolute value was different (Figure 5E). The expression level of the aggressive cancer-related genes was similar to the log2 FC of the DEGs (Figure 5F). The expression changes of the mitogen-activated protein kinase (MAPK) and the nuclear factor kappa B (NF-κB) pathway-related proteins as major sub-mechanisms of the TNF were confirmed (Figure 5G,I). The results showed that there was no significant change in the expression of extracellular signal-regulated protein kinases (ERK)1/2 as a sub-protein of the MAPK pathway. The expression of p105 as a precursor of NF-κB was clearly decreased, and the expression of p50 as an active form of NF-κB was slightly increased. These results indicated that the activity ratio of NF-κB was significantly increased. The results proved that the formation of PGCCs by fludioxonil was through the NF-κB pathway as sub-mechanisms of the TNF pathway and the cytokine-related pathway.

### 2.5. Characteristics of the Daughter Cells Formed by Exposure to Fludioxonil

The mother cell, directly exposed to fludioxonil, was incubated at low density for 7 days. A small colony (yellow arrow and line) was formed around the mother cells (PGCCs-Flu) and subcultured at low density using 1% trypsin-EDTA. This process was repeated three times, separating the daughter cells from the mother cells. The mother cells were not detached by the treatment with 1% trypsin-EDTA (Figure 6A). The morphology of the daughter cells showed the enlarged cell body similar to the mother cells (Figure 6B). The daughter cells compared with the mother cells showed that the polyploid nuclei were not maintained. However, the size of the nuclei in the cells was significantly more enlarged than the control (Figure 6C).

The migration ability of the daughter cells was still superior to the control group and similar to that of the mother cells (Figure 7A,B). In addition, the resistance to the three types of anticancer drugs was also significantly dominant compared to the control group. However, the difference from the control group was lower compared to that of the mother cells (Figure 7C–E). The expression of the cell cycle-promoting proteins (Cyclin D1, Cyclin E1, and Cyclin B1) was increased compared to the control group, and the expression of p53 was also maintained at an elevated level (Figure 8A–D). The expression of p50, an active form of NF-κB, was significantly increased, and the expression of the NF-κB protein was highly maintained, as in the mother cells (Figure 8C–E). The expression of E-cadherin was slightly decreased, and that of Vimentin was significantly increased, supporting the increase in the migration ability. The expression of SOX2 was significantly increased as in the mother cells (Figure 8F,G).

## 3. Materials and Methods

### 3.1. Chemicals

Fludioxonil (CAS No. 131341-86-1), 17β-estradiol (E2; CAS No. 50-28-2), doxorubicin (CAS No. 25316-40-9), 5-fluorouracil (5-FU; CAS No. 51-21-8), cisplatin (CAS No. 15663-27-1), anhydrous dimethyl sulfoxide (DMSO; CAS No. 67-68-5), and *N,N*-dimethylformamide (DMF; CAS No. 68-12-2) were purchased from Sigma-Aldrich (St. Louis, MO, USA). The stock solution (excluding cisplatin) was prepared in anhydrous DMSO (Sigma-Aldrich), and the stock solution of cisplatin was prepared in DMF (Sigma-Aldrich) at 1000 times the concentration of the working solution and kept at 4 °C temperature until required. The working solution of chemicals (excluding cisplatin) was prepared in anhydrous DMSO (Sigma-Aldrich) and kept at room temperature. The working solution of cisplatin was left at room temperature before proceeding. The concentration of 0.1% DMSO was selected at the same vehicle concentrations when treated with fludioxonil and defined as a control.

### 3.2. Cell Media, Culture, and Seeding

MDA-MB-231, T47D, and MCF-7 were the well-known human breast cancer cell lines. Each human breast cancer cell line has different cancer characteristics (Table 1). MDA-MB-231, T47D, and MCF-7 breast cancer cells were purchased from the Korean Cell Line Bank (Jongno-gu, Seoul, Republic of Korea) for non-profit purposes and cultured using Media based on Dulbecco’s Modified Eagle Media (DMEM; Fischer Scientific, Hampton, NH, USA) at 37 °C in a humidified CO_2_ incubator.

The media included fetal bovine serum (FBS; R&D Systems, Minneapolis, MN, USA). FBS consists of cell growth factors, steroid hormones, and nutrients to grow the cell that can hinder the net effect of chemicals in experiments [28]. For this reason, we used charcoal–dextran stripped FBS (CD-FBS) in phenol-free low glucose DMEM (p/f DMEM) (Welgene Inc., Gyeongsan, Republic of Korea) (Table 2) in ‘phenol-free media’. The dextran-coated charcoal was purchased from Sigma-Aldrich and mixed with FBS (R&D systems) following processing methods.

**Table 1 ijms-25-09024-t001:** Character of breast cancer cell lines [29,30,31].

Cell Lines	ER	PR	HER2	BRCA1 Mutation	Subtype	Tumor Type	p53 Gene
MDA-MB-231	−	−	−	WT	TNB	AC	Mut
T47D	+	+	−	WT	LA	IDC	Mut
MCF-7	+	+	−	WT	LA	IDC	WT

WT: wild type, Mut: mutation, TNB: triple-negative breast cancer, LA: luminal A, AC: adenocarcinoma, IDC: invasion ductal carcinoma.

Cells were separated using 0.05% trypsin-EDTA (Life Technologies Inc., Carlsbad, CA, USA) for 2 min at 37 °C in a humidified CO_2_ incubator after reaching about 90% of the culture dish surface and then subcultured at a split ratio of 1:10 (MDA-MB-231). Media were replaced every 3 days.

Three breast cancer cell lines were seeded in suitable plates containing ‘cell seeding media’ and stabilized for 24 h. After that, cells were starved for 24 h by incubating in ‘phenol-free media’ before chemical treatment (starvation process). After starvation, cells were incubated at 37 °C in a humidified CO_2_ incubator in ‘phenol-free media’ containing the appropriate chemicals following the experiment schedule (Table 3).

### 3.3. WST Assay for Identification of Cell Viability

The WST assay was performed to evaluate cell viability. Briefly, MDA-MB-231, T47D, and MCF-7 breast cancer cells were seeded at a density of 2 × 10^3^ cells/well, respectively, in clear flat-bottom 96-well plates (SARSTEDT AG & Co., Nümbrecht, Germany), and cultured at 37 °C in a humidified CO_2_ incubator. The cells were cultured according to the treatment schedules (Table 3). All media were subsequently removed, and Quanti-MaxTM WST-8 Cell Viability Assay (BIOMAX, Gyeonggi, Republic of Korea), which produces the WST, was added to each well and incubated for 1 h at 37 °C in a humidified CO_2_ incubator. Absorbance was subsequently measured at 450 nm using a multi-mode microplate reader (Synergy Neo2, BioTek Instruments Inc., Winooski, VT, USA).

### 3.4. Morphology Staining

The staining of the cell was performed to confirm the morphology change by treatment of chemicals. In short, MDA-MB-231 and MCF-7 cells were seeded at a density of 2.0 × 10^4^ cells in an 8-well cell culture slide (PS/Flux/PP) (SPL Life Sciences Co., Gyeonggi, Republic of Korea) and incubated at 37 °C in a humidified CO_2_ incubator. The seeded cells were treated with the chemicals for 72 h described in Table 3. Treated cells were subsequently fixed in 4% paraformaldehyde (Sigma-Aldrich) for 10 min at 37 °C and permeabilized in 0.1% Triton X-100 (Sigma) in DPBS for 15 min at room temperature. In cell body staining, MDA-MB-231 cells were dyed with 5 µg/mL Calcein, AM cell-permeant dye (Invitrogen, Waltham, MA, USA, Cat No. C1430), and 0.2 mg/mL bisBenzimide H33342 trihydrochloride (Hoechst, Sigma-Aldrich) in DPBS for 15 min at room temperature. In actin staining, MDA-MB-231 cells were dyed with ActinGreen 488 Readyprobes Reagent (Invitrogen) and 0.2 mg/mL Hoechst (Sigma-Aldrich) in DPBS for 15 min at room temperature. After washing with DPBS, the samples were sealed with the mounting medium (ImmunoBioScience Corp, Mulkilteo, WA, USA, Cat No. AR-6515-01). The slides were examined using the IX-73 inverted microscope (Olympus, Tokyo, Japan). The Gen5 image analysis software (version 3.12, BioTek Instruments Inc.) was used to quantify the number of cells, cell nuclei, and body size.

### 3.5. Live-Cell Image System for Identification of Cell Division

Live-cell imaging was performed to confirm cell division in real time for 84 h immediately after treatment with 0.1% DMSO or fludioxonil. Briefly, MDA-MB-231 cells were seeded and cultured at 37 °C at a density of 1.5 × 10^4^ cells/well in transparent flat-bottom 6-well plates (SARSTEDT AG & Co.). After the ‘starvation process’, the seeded cells were treated with 0.1% DMSO or fludioxonil and recorded at 30 min intervals for 84 h using Lionheart FX Live Cell Imaging System (BioTek Instruments Inc.). The media, included the chemicals, were replaced at 48 h. The images were edited into the video using Gen5 image analysis software (BioTek Instruments Inc.) and Movavi Video Editor 2023 (version 22.4.1., Movavi Software Inc., Wildwood, MO, USA).

### 3.6. Cell Cycle Analysis Using the FACS

Cell cycle analysis was conducted to measure the percentage of MDA-MB-231 breast cancer cells in each cell cycle phase. Briefly, cells were seeded at a density of 1.0 × 10^5^ cells of MDA-MB-231, respectively, in 90 mm cell culture dishes (SPL Life Sciences Co., Ltd.). Seeded cells were treated as per the protocol mentioned in Table 3, for 72 h. Treated cells were counted for the cell number after trypsinization and adjusted at 2.0 × 10^6^ cells/mL. The cell suspensions were slowly and gently added to cold 70% ethanol and stored for 2 h at 4 °C. After the centrifugation, the cell suspensions were stained with 1.0 mg/mL propidium iodide (PI) solution (Sigma-Aldrich) containing 10 mg/mL RNase A solution (Sigma-Aldrich), overnight at 4 °C. Cell suspensions were filtered using a 70 μm pore size cell strainer to prevent cell aggregation. Each phase of the cell cycle was analyzed using FACS (fluorescence-activated cell sorting system; Sony SH800 Cell Sorter, Tokyo, Japan), and calculated by applying the ModFit LT 5.0 program (Verity Software House, Topsham, ME, USA).

### 3.7. Scratch Assay to Evaluate Cell Migration

The scratch assay was conducted to evaluate cell migration ability for 48 h in viable cells. Briefly, cells were seeded at a density of 1.0 × 10^5^ cells of MDA-MB-231, respectively, in 90 mm cell culture dishes (SPL Life Sciences Co., Ltd.). Seeded cells were treated with the chemicals for 72 h. Treated cells were counted for the cell number after trypsinization and seeded at 2.0 × 10^5^ cells/well (0.1% DMSO treatment group) or 1.5 × 10^5^ cells/well (fludioxonil 10^−5^ M treatment group) in 6-well plates (SARSTEDT AG & Co.). After stabilization for 24 h, seeded cells were treated with 25 μL/well mitomycin C (Sigma-Aldrich, 12.5 μg/mL final concentration in media) for 2 h, in a 37 °C humidified CO_2_ incubator. Subsequently, each seeded well was scratched using a sterile 1 mL pipette tip and washed twice using Dulbecco’s Phosphate-Buffered Saline (DPBS, Welgene Inc.). The scratched wells were immediately filled with the ‘chemical treatment media’ and photographed using the IX-73 inverted microscope (Olympus) at 0 h. The plate was incubated for 48 h and photographed as per the protocol mentioned in Table 3. The scratch area was quantified by applying the CellSens Dimension 1.13 version (Olympus).

### 3.8. Evaluation of Chemo-Resistance

The chemo-resistance was evaluated using the WST assay to confirm the cell viability by treatment with chemotherapy medication to treat cancer. The MDA-MB-231 breast cancer cells were seeded at a density of 1.0 × 10^5^ cells of MDA-MB-231, respectively, in 90 mm cell culture dishes (SPL Life Sciences Co., Ltd.). Seeded cells were treated with the chemicals for 72 h. Treated cells were counted for the cell number after trypsinization and seeded at 2.0 × 10^4^ cells/well in 24-well plates (SARSTEDT AG & Co.). After 24 h, seeded cells were treated with doxorubicin, 5-FU, and cisplatin for 72 h. All media were subsequently removed, and Quanti-MaxTM WST-8 Cell Viability Assay (BIOMAX) was added to each well and incubated for 1 h at 37 °C in a humidified CO_2_ incubator. Absorbance was subsequently measured at 450 nm using a multi-mode microplate reader (BioTek Instruments Inc.).

### 3.9. mRNA Sequencing and Analyses

To confirm the change of messenger RNA (mRNA) by treatment with fludioxonil (10^−5^ M), mRNA sequencing was conducted and analyzed. MDA-MB-231 breast cancer cells were seeded at a density of 1.0 × 10^5^ cells in 90 mm cell culture dishes (SPL Life Sciences Co., Ltd.). Seeded cells were treated with the chemicals for 72 h. After trypsinization, the cell pellets were resuspended using the RNAlater Solution (Invitrogen). The mRNA sequencing and differentially expressed gene (DEG) analysis were entrusted to DNALink Co., Seoul, Republic of Korea.

mRNA sequencing using TruSeq Stranded mRNA/total Library Prep kit was prepared in the following manner. RNA purity of the sample was confirmed by assaying 1 μL of total RNA extract on a NanoDrop8000 spectrophotometer (Thermo Fisher Scientific Inc., Waltham, MA, USA). Total RNA integrity was checked using a 2100 Expert Bioanalyzer (Agilent, Santa Clara, CA, USA) with an RNA Integrity Number (RIN) value. mRNA sequencing libraries were prepared according to the manufacturer’s instructions (TruSeq Stranded mRNA/total Library Prep kit). mRNA samples were purified and fragmented from total RNA (1 μg) using poly-T-oligo-attached magnetic beads on two-times purification. Cleaved RNA fragments primed with random hexamers were reverse-transcribed into first-strand complementary DNA (cDNA) using reverse transcriptase, random primers, and 2′-deoxyuridine, 5′-triphosphate (dUTP). These cDNA fragments were connected by adding a single ‘A’ base and subsequent adapter ligation. The products were enriched after polymerase chain reaction (PCR) purification to create the final strand-specific cDNA library. The quality of the final cDNA was confirmed by capillary electrophoresis (2100 Bioanalyzer instrument, Agilent). After conducting quantitative PCR using SYBR Green PCR Master Mix (Applied Biosystems, Waltham, MA, USA), libraries were combined with the index tagged in equimolar amounts in the pool. Cluster generation occurred in the flow cell on the cBot automated cluster generation system (Illumina Technologies, San Diego, CA, USA). The flow cells were then loaded on Novaseq 6000 sequencing system (Illumina Technologies) and sequencing was conducted with 2 × 100 bp read length.

DEG analysis was conducted as mapping reads on a reference genome and calculating expression between samples. At first, reads for each sample were mapped to the reference genome by Tophat (version 2.0.13, the program arranged RNA-Seq reads to confirm the exon–exon splice junction, Johns Hopkins University). The aligned results were added to Cuffdiff (version 2.2.0, the program of transcriptome assembly and differential expression analysis for RNA-Seq) to report differentially expressed genes. To normalize the library and estimate dispersion, geometric and pooled (“blind” was used in single replicates, or “pooled” was used in multiple replicates) methods were applied.

Identification of DEGs and functional enrichment analysis was conducted in the following manner. The “gene_exp.diff”, the output files of Cuffdiff, was used to identify the DEGs. Two filtering processes were applied to detect DEGs among the sample treated with 0.1% DMSO or Fludioxonil. First, genes with only “OK” status were extracted using the Cuffdiff status code. The status code means condition of enough reads in a reliable calculation of expression level. And “OK” status means the successful calculating of the gene expression level. Biologically significant DEGs were characterized as genes exhibiting a statistically significant difference in expression using edgeR’s default method to adjust for false discovery rate (FDR ≤ 0.05) and fold change (FC) greater than 2 in either the positive or negative direction (Appendix A). The graphs were made based on the Gene Ontology (GO) biological process, GO molecular function, and Kyoto Encyclopedia of Genes and Genomes (KEGG) pathway using program tool ShinyGO version 0.77 (South Dakota State University, Brookings, SD, USA).

### 3.10. Quantitative Reverse Transcription Polymerase Chain Reaction (qPCR)

qPCR was conducted to confirm the expression levels of significantly changed genes in mRNA sequencing data. MDA-MB-231 cells were cultured following the schedule of Table 3 to obtain the RNA. Total RNA was obtained from cells using the TRIzol reagent (Invitrogen) following the manufacturer’s protocol. The RNA pellet was dissolved in DEPC-treated distilled water (Invitrogen), and total RNA concentration was measured using the microreader (Take 3, BioTek Instruments Inc.) at 260/280 nm. cDNA synthesis was conducted using the PrimeScript RT Master Mix (Takara Bio Inc., Kusatsu, Japan) following the manufacturer’s protocol. The cDNA was amplified using the each primers as shown in Table 4 (Bioneer Co., Daejeon, Korea), and the TB Green Premix Ex Taw Ⅱ (Takara Bio Inc.). qPCR was operated for 40 cycles comprising 95 °C for 15 s, 60 °C for 30 s, and 72 °C for 30 s using the QuantStudio 3 Real-Time PCR System (Applied Biosystems). The housekeeping gene 18 s was used as the internal control. The expression of each sample was calculated using the following formula:∆Ct = Ct (target gene) − Ct (18 s-rRNA)
∆∆Ct = ∆Ct (treatment group) − ∆Ct (vehicle control)
∴ 2 ^−∆∆Ct^ (final value)

### 3.11. Western Blot Analysis

Western blot analysis was conducted to verify the expression levels of cell cycle-related, migration, and CSCs-related proteins using ProteinSimple JESS (Bio-Techne, Miniapolis, MN, USA). MDA-MB-231 breast cancer cells were seeded at a density of 1.0 × 10^5^ cells in 90 mm cell culture dishes (SPL Life Sciences Co., Ltd.). Seeded cells were treated with the chemicals for 72 h. Treated cells were lysed overnight by RIPA lysis buffer (ATTO, Tokyo, Japan) containing protease inhibitor (ATTO). Each protein concentration was confirmed using the bicinchoninic acid (Sigma-Aldrich Inc.) assay, and average values of tri-repeat were determined. After the adjustment of protein concentration at 2.5 μg/3 μL in the well, the cell lysates were analyzed following the manufacturer’s methods (12–230 kDa Separation Module, ProteinSimple of Bio-Techne). For the stable reaction of the antibody binding, the conditions of analysis were adjusted as follows: separation time 35 min, separation voltage 375 volts, blocking time 60 min, primary antibody time 90 min, and secondary antibody time 75 min. The final value was the peak area excluding the base level of the chromatograph and obtained from Compass for SW (v 6.2.0). All protein expression levels (Table 5) in each band are adjusted to GAPDH protein expression levels. In addition, the treatment groups of each batch were calculated as a ratio of the control groups of the same batch. The final ratio value was presented in the graph using GraphPad Prism 5.01 software (GraphPad Software Inc., San Diego, CA, USA).

### 3.12. Acquisition of Daughter Cells

The daughter cells were obtained from the small colony, that was emerged after the PGCCs-treated fludioxonil 10^−5^ M for 72 h on MDA-MB-231 breast cancer cells. These cells related to the poor prognosis in cancer metastasis [3].

MDA-MB-231 breast cancer cells were seeded at a density of 1.0 × 10^5^ cells in 90 mm cell culture dishes (SPL Life Sciences Co., Ltd.). Seeded cells were treated as per the protocol mentioned in Table 3, for 72 h. Treated cells were counted for the cell number after trypsinization and adjusted at 1.0 × 10^5^ cells in 90 mm cell culture dishes (SPL Life Sciences Co., Ltd.). After 8 days, the cells were trypsinized, counted, and subcultured in new cell culture dishes. When the cells were subcultured after being treated with fludioxonil 10^−5^ M for 72 h, cells were dissociated using 0.1% trypsin-EDTA (Life Technologies Inc.) for 2 min at 37 °C in a humidified CO_2_ incubator. The media used were ‘phenol-free media’ and freshly changed every 2–3 days. After the 3rd subculture, the daughter cells were obtained as isolation like a cell line, that was used a diverse experiment to exam the poor prognosis in cancer metastasis. The daughter cells were each obtained by 4 repetitions of the independent process. The morphology staining, scratch assay, test of chemo-resistance, and Western blot of the daughter cells were conducted as previously described without the treatment of chemicals.

### 3.13. Statistical Analysis

All experiments were performed at least three independent times, and all data have been presented as means ± standard error of the mean (SEM). Statistical analysis was determined by one-way analysis of variance (ANOVA) followed Dunnett’s multiple comparison, *t*-test (paired test), or two-way ANOVA followed by Bonferroni posttests using the GraphPad Prism version 5.01 software (GraphPad Software Inc.). A *p* < 0.05 was considered statistically significant. *: *p* < 0.05 compared to control.

## 4. Discussion

The use of pesticides is essential for protecting crops and preventing and mitigating harmful pests (insects, mice, other animals, weeds, fungi, etc.) for an abundant harvest of agricultural products in limited arable land. Pesticides are also used to prevent serious diseases transmitted from harmful microorganisms such as bacteria and viruses, in hospital and medical environments. To tend to gardens or grass plots, we have no choice but to use pesticides. Thus, the use of pesticides is unavoidable.

The wide use of pesticides affects human and environmental health. Exposure to pesticides may occur through multiple routes, such as oral, dermal, and inhalation, depending on the conditions or the type of pesticide used. Typically, consumers may be exposed to pesticide residues by consuming food (fruits, vegetables, grains, and other agricultural products) and drinking water [32]. These routes serve as a major part of the food chain. Some pesticides are resistant to breakdown, resulting in bioaccumulation through the food chain. Pesticides are lipophilic and tend to concentrate in the fatty tissues of animals that humans may consume [33]. Water molecules carry away soluble pesticides to reach surface water and groundwater. Pesticides do not decompose in groundwater due to the lack of light and oxygen. For this reason, pesticide spraying degrades water quality and reduces the quality of potable water. The accumulation of pesticides by long-time exposure at low concentrations [34] is linked to a wide range of diseases caused due to endocrine disruption, and carcinogenic, hepatotoxic, cytotoxic, and neurotoxic effects, leading to reproductive problems, and teratogenicity and abnormal fetal development, among many others [35].

In an earlier study, multiple pesticides present in mothers affected the weight, size, and head circumference of the baby at birth. In addition, breastfed infants are known to acquire pesticides from the mother [36]. Bioaccumulating pesticides are found in breastfed infants at levels higher than the Food and Agriculture Organization (FAO)/World Health Organization (WHO) permissible tolerable daily intake values [34,35]. To know the dangers and minimize the adverse side effects of pesticides, regular monitoring and research appeared to be required [33].

Fludioxonil is used worldwide but is classified as very toxic to aquatic life with long-lasting effects following European Chemicals Agency (ECHA, last updated: 17 August 2024). By the Globally Harmonized System (GHS) classification of the Japanese government (NITE), fludioxonil was classified in category 2 of carcinogenicity in 2019. Nevertheless, fludioxonil’s effects on carcinogenesis or cancer development have not been studied in depth to date.

The current study was conducted to establish the effect of fludioxonil on the formation of PGCCs, which leads to a poor prognosis in TNBC. In the cell viability assay, the viability of fludioxonil-treated MCF-7 and T47D cells was similar to that of E2 (ERα agonist). Furthermore, the morphology of the MCF-7 cells was not altered by fludioxonil treatment. However, in MDA-MB-231 cells, the fludioxonil treatment resulted in a reduction in the cell viability without cell apoptosis and changed morphology, with an increase in the number of nuclei in the cells, enlarged cell nuclei, and cell bodies. In live-cell imaging, fludioxonil exposure resulted in the transformation of cell morphology with prolonged mitotic time and the presence of more than two nuclei. The cell cycle analysis revealed that fludioxonil exposure induced a new peak besides the G0/G1, S-, and G2/M phases. The new peak was identified as PGCCs having over two nuclei in a cell, which was confirmed after 24 h of fludioxonil exposure. At 72 h, the center of the new peak migrated to the higher value direction, and the peak ratio in the cell cycle was higher than that at 24 h. Cyclin E1 and p53 expression were significantly increased in this process, despite the p53-mutated TNBC cell line.

These results are supported by many previous studies. Mutations of the p53 tumor suppressor gene (mutp53) appear in more than 50% of all human cancers and are commonly expressed at high levels compared to the wild-type p53 gene. This reflects a positive selection as the missense mutation of p53 differs from other tumor suppressor genes such as truncation or deletion mutants [37]. The expression of the mutp53 isoform is another significant factor in cancer development and is known to promote transformation through oncogenic gain-of-function (GOF) activities. This may be related to its enhanced invasion properties, inhibiting apoptosis, and increasing genomic instability [38,39,40]. The cells with mutp53 have been regulated to have higher tumorigenic properties than the cells of wild-type p53 [41]. The MDA-MB-231 breast cancer cells are typically of the mutp53 gene TNBC breast cancer cell line, whereas MCF-7 has wild-type p53. However, MDA-MB-231 cells express very high levels of p53 relative to MCF-7 cells. It has been documented that the MDA-MB-231 cells are able to show various p53-dependent reactions [42].

The multinucleated cell (included polyploid and aneuploidy nucleated cell) in tumors is closely related to p53. Whole-genome duplication (WGD), a frequent event in cancer evolution and an essential driver of aneuploidy, was regulated by p53. This role is enigmatic as can be seen from the following: p53 can block the proliferation of polyploid cells, acting as a blocker of WGD, but can also promote replication without mitosis, which is a key step in WGD via endo-reduplication [43]. Another study revealed that WGD via endo-reduplication after telomere attrition occurs only in cells lacking p53 [44]. The p53-mutated lymphoma cells have been studied for the formation of PGCCs subsequent to the DNA damage that occurs after chemo- and radiation therapy. After severe DNA damage, the cells undergoing aberrant mitosis are arrested at the G2/M phase of the cell cycle checkpoint. Subsequently, the cells enter the endo-reduplication cycle, forming PGCCs. When the PGCCs are incubated after the isolation, they produced the new cells as clonogenic survivors [45]. These studies support the hypothesis that fludioxonil can induce the development of PGCCs in p53-mutant TNBC MDA-MB-231.

Cyclin E1 is a highly conserved cell cycle regulation protein that activates Cyclin-dependent kinase 2 to promote cell cycle progression. Cyclin E1 plays a major role in the normal cell cycle and the normal development of the polyploid giant trophoblasts in the placenta. However, the overexpression of Cyclin E1 could trigger oncogenicity. Cyclin E1 is overexpressed in multiple malignancies that correlate with the survival of cells. Moreover, transgenic mice with deregulated Cyclin E1 expression developed carcinomas. Similarly, Cyclin E1 exhibits oncogenic properties with the ability to increase cell proliferation and induce genomic instability. An earlier study proved that the overexpression of Cyclin E1 inhibits the binding of anaphase-promoting complex and cdc20 homolog, leading to delayed mitosis through a delay in the degradation of Cyclin B1. It induced mitotic slippage and resulted in a 4N (N: sets of chromosomes) nucleus in a cell [46,47,48,49,50,51].

Furthermore, Cyclin E is particularly associated with polyploidy in high-grade serous ovarian cancer [52]. In another study, the elevated expression of Cyclin E1 induced replicative stress and p53-dependent mitotic bypass. p53 acts via p21 and promotes mitotic bypass by inhibiting Cyclin-dependent kinase. The elevated expression of Cyclin E drives the complete endo-reduplication via p53-G1-arrested cells. Consequently, Cyclin E can induce WGD, which can divide the polyploidy in p53-fully-expressed cells [43]. These previous studies supported the results of this study that Cyclin E1 is more expressed than other cell cycle-related proteins in breast cancer PGCCs, and it is predicted to be related to the expression of p53.

In the results of the NGS, the genes present in the PGCCs formed by fludioxonil exposure were analyzed for the up-regulation of ‘cell population proliferation’ and ‘cell death’ by an analysis of the GO biological process. Based on the KEGG pathway, the genes were changed by the increase in ‘cytokine–cytokine receptor interaction’, which was supported by the Tree diagram of the GO biological process and the KEGG pathway, relating to have a correlation with the TNF and NF-κB signaling pathways. Genes related to TNF signaling suggested in the DEGs were confirmed with the results of qPCR, and the expression tendencies of the genes were remarkably similar. NF-κB was confirmed to be more activated than MAPK through the protein expression. Based on these results, the PGCCs formed by fludioxonil exposure were believed to be formed via the NF-κB signaling pathway, including the activation of cytokine-related action.

Inflammation is known to be closely linked with cancer, but a mechanistic comprehension of this complex relationship is difficult. In tumors, inflammatory chemokines and cytokines such as TNF regulate the activity of infiltrating immune cells, and the behavior of cancer cells influences tumor growth, dissemination, and response to therapy [50]. Various studies suggest that the oncogenic properties of mutp53 are deeply connected to the inflammatory environment of tumors. An earlier study using a mouse model of inflammatory bowel disease reported that mutp53 stimulates the pro-inflammatory microenvironment. In addition, mutp53 may have an unexplored role in the response to the inflammation of already-transformed cancer cells. When an inflammatory microenvironment has been established within the growing tumor mass, a large proportion of tumors go through p53 mutation at a later stage of transformation [53,54,55,56]. In addition, in the inflammatory tumor microenvironment, TNF stimulates NF-κB signaling in mutp53 cells, leading to invasion, metastasis, and increased lymphocyte infiltration. Consequentially, the cells obtain their aggressive ability via GOF mutp53 [56].

Activating NF-κB as a transcription factor for inflammatory response promotes cell proliferation, apoptosis, migration, invasion, and angiogenesis during tumor development. Moreover, it induces genetic instability and epigenetic modifications [57,58]. The activation of NF-κB in cancer cells resulted in the induction of CSCs, EMT, and resistance to chemotherapy. NF-κB can control inflammatory responses in the tumor microenvironment, which supports tumor initiation and progression [59,60,61,62,63,64]. Specifically, mutp53 has been known to increase NF-κB activation in cells, presumably through direct protein–protein interaction [65,66,67]. Activation of NF-κB is known to relate with the diverse action by the GOF of mutp53 in cells [39,68,69]. The earlier study showed that PGCCs can induce cytokine-mediated signaling, including intense pro-inflammatory signaling (interleukin [IL]-1B, IL-6, C-C motif chemokine ligand 4 like 2 [CCL4L2], and C-X-C motif chemokine ligand 10 [CCL10]) by polyploid cells. The cytosolic DNA produced by mitotic slippage was partially protected from the innate and inflammatory immune response, particularly at the beginning [70]. These studies support the results of this study regarding the relation between the PGCCs formed due to fludioxonil exposure and the inflammatory responses via the NF-κB pathway in mutp53 TNBC cells.

After the formation of PGCCs by exposure to fludioxonil, an increase in migration ability and chemo-resistance was observed in this study. Also, the expression of EMT (decrease in E-cadherin, increase in Vimentin) and CSC-related proteins (Nanog and SOX2) were up-regulated compared with the control. The daughter cells were incubated after separating from the colony around the mother cells (PGCCs-Flu) for a long time. The production of daughter cells occurred in some mother cells, and the process was undeterminable. The daughter cells maintained the hetero-morphology, significant migration ability, and chemo-resistance, although they displayed a single nucleus in a cell with a decreased cell body size compared with the mother cells. Cyclin E1, p53, and NF-κB were highly expressed in the mother cells (PGCCs-Flu), and Vimentin and SOX2 were also higher expressed compared with the control.

The PGCCs and the daughter cells exhibited CSC-like features. The evidence of PGCCs with stem cell characteristics is suggested for three reasons: strong proliferation ability, differentiation into some normal tissues, and expression of stem cell markers. In previous studies, the PGCC had stem cell-like properties and an ingenious proliferation cycle, like an embryonic development cycle [71,72,73]. Traditionally, because the mitotic cell cycle was considered the main mode by which eukaryotic cells divided, PGCCs were considered only a side product of tumor progression, nonviable, and not a potential cause of tumorigenesis [74]. However, it has been gradually realized that PGCCs can actually release daughter cells via the ‘female pregnancy-like system’, a more primitive amitotic and budding division rather than the traditional mitotic division [70,74]. In another study, sorting single multinucleated tumor cells injected under the skin of mice led to a successfully constructed PGCC level in vivo tumor model. In addition, PGCCs have been induced to differentiate in special media into several mesenchymal tissues, such as adipose, cartilage, and bone [6]. Based on some earlier studies, the PGCCs can be differentiated into erythrocyte-like cells and embryonic hemoglobin-like cells with oxygen-binding ability. This ability improves the hypoxic environment inside the tumor [75,76]. In addition, the PGCCs highly express the CSC markers such as OCT4, SOX2, and Nanog, which suggest multi-differential potential and self-renewal ability like an embryonic stem cell [77,78]. The PGCCs and daughter cells have been observed to form a cluster with more powerful invasion and migration abilities, accompanied by N-cadherin, Vimentin, Twist, Slug, Snail, and Cathepsin [75,79,80]. In another study, the PGCCs subcultured 10 times have been shown to have more vigorous invasion and metastasis ability than the primary generation, even untreated control cells [81].

Fludioxonil, used worldwide in agriculture, accumulates Cyclin E1 and promotes the inflammatory cytokine-enriched microenvironment via the up-regulation of TNF and NF-κB. This has led to the transformation of cancer cells to PGCCs via abnormal cell cycles, such as the mitotic delay and mitotic slippage in mutp53 TNBC MDA-MB-231 cells. The PGCCs and the daughter cells had significant migration ability, chemo-resistance, and cancer stemness (Figure 9). The results of this study strongly suggest that fludioxonil, as an inducer of the potential genotoxicity, induces the formation of the PGCCs, leading to the formation of metastatic and stem cell-like breast cancer cells.

### Limitations of the Study

This study has the following limitations: lack of confirmation of the formation of PGCCs in animal models, too high fludioxonil concentration compared with actually exposed concentration in humans, and less diversity of the cell lines.

## Figures and Tables

**Figure 1 ijms-25-09024-f001:**
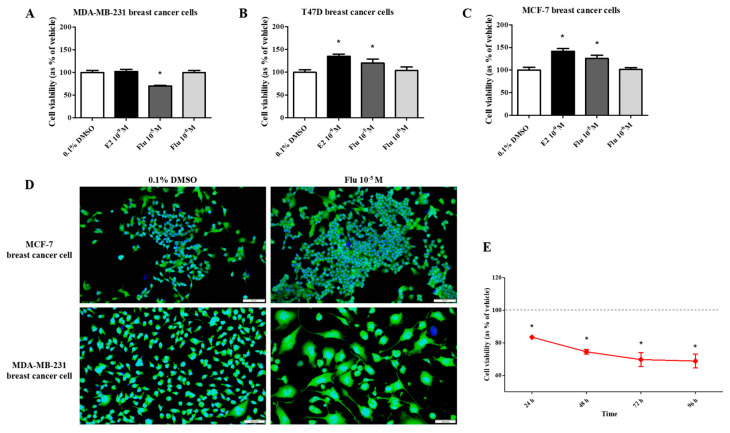
Effect of cell viability by applying the WST assay and DAPI and actin staining after treatment with fludioxonil in MDA-MB-231, T-47D, and MCF-7 breast cancer cells. The cell viability assay was conducted after treatment with fludioxonil for 72 h in (**A**) MDA-MB-231, (**B**) T-47D, and (**C**) MCF-7 breast cancer cells. (**D**) The image presents the difference in cell viability by treatment with fludioxonil for 72 h in MCF-7 and MDA-MB-231 breast cancer cells. (**E**) The cell viability of MDA-MB-231 breast cancer cells by treatment with fludioxonil was changed in a time-dependent manner. Flu: fludioxonil; E2: 17β-estradiol. DMSO was used as the vehicle control, and the value of the control containing 0.1% DMSO was set as 100%. Data in the graphs are obtained from at least three repeated experiments and are presented as the mean ± SEM. Statistical analysis was determined by (**A**–**C**) one-way ANOVA followed by Dunnett’s multiple comparison and (**E**) two-way ANOVA followed by the Bonferroni posttests using the GraphPad Prism 5.01 software. *p* < 0.05 was considered statistically significant. *: *p* < 0.05 compared to control. Scale bar = 100 μm or 200 pixels.

**Figure 2 ijms-25-09024-f002:**
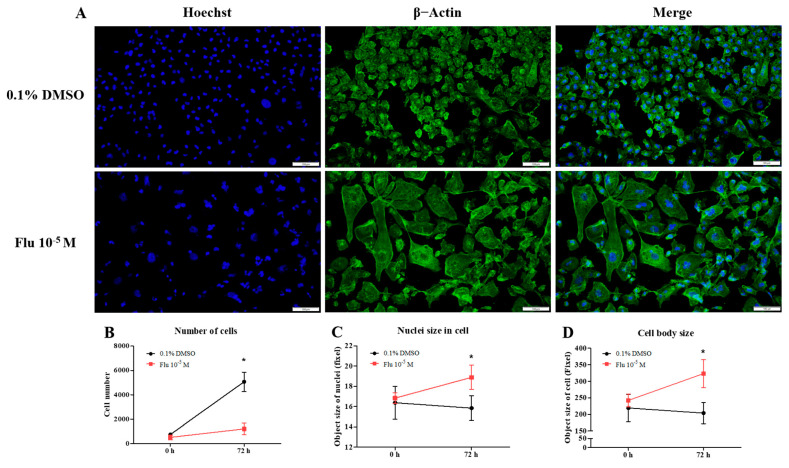
Change of cell morphology after treatment with fludioxonil in MDA-MB-231 breast cancer cells. (**A**) The cell morphology of MDA-MB-231 breast cancer cells after they were treated with fludioxonil for 72 h and were stained with DAPI and actin. These images were analyzed to count (**B**) nuclei number and to measure the (**C**) size of nuclei and (**D**) cell body. Flu: fludioxonil; E2: 17β-estradiol. DMSO was used as the vehicle control, and the value of the control containing 0.1% DMSO was set as 100%. Data in the graphs are obtained from at least three repeated experiments and are presented as the mean ± SEM. Statistical analysis was determined by (**B**–**D**) two-way ANOVA followed by Bonferroni posttests using the GraphPad Prism 5.01 software. *: *p* < 0.05 compared to control. Scale bar = 100 μm or 200 pixels.

**Figure 3 ijms-25-09024-f003:**
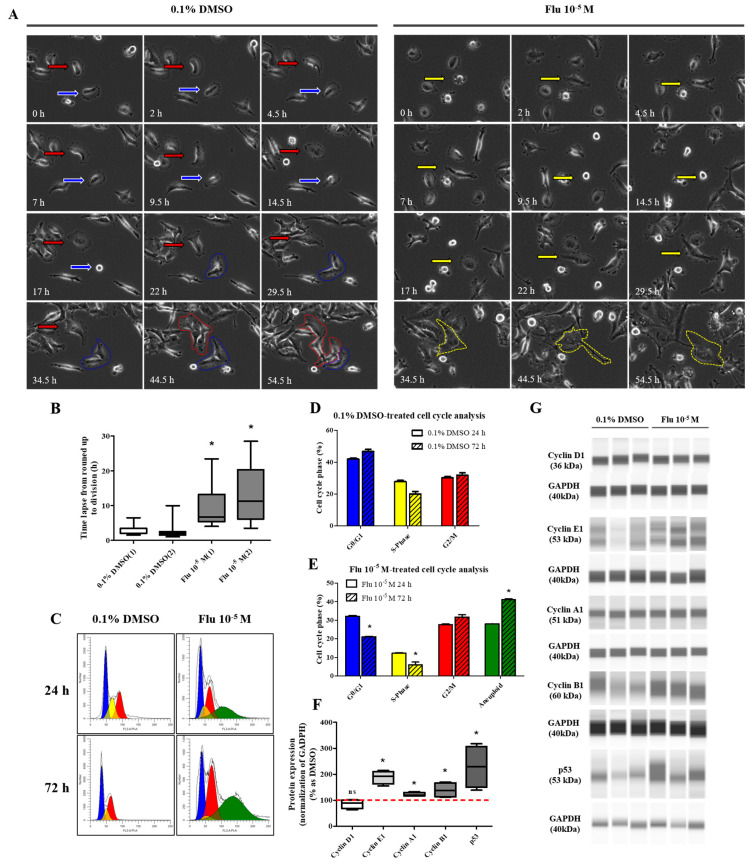
Display of live-cell images in time-dependent manner, as well as the change in cell-cycle and related proteins of MDA-MB-231 breast cancer cells after exposed of 0.1% DMSO or fludioxonil. (**A**) The cells after seeding were treated with fludioxonil for 72 h. During that time, the real-time videos of cells were recorded using BioTek Lionheart FX automated microscope. Blue arrow and line: normal cell division. Red arrow and line: temporarily abnormal cell division. Yellow arrow and line: production of PGCC. (**B**) Based on these videos, the time lapse from cells’ round-up to division was counted and displayed on the graph. (**C**) The cells were analyzed for the cell cycle after being treated with the 0.1% DMSO or fludioxonil for 72 h, which were displayed in the graph (**D**,**E**). The change of cell cycle-related proteins was analyzed using Jess from Protein Simple (**F**,**G**). Flu: fludioxonil. DMSO was used as the vehicle control, and the value of the control containing 0.1% DMSO was set as 100%. Data in the graphs are obtained from at least three repeated experiments and are presented as the mean ± SEM. Statistical analysis was determined by (**B**) one-way ANOVA followed by Dunnett’s multiple comparison, (**D**,**E**) two-way ANOVA followed by Bonferroni posttests, and (**F**) *t*-test (paired test) using the GraphPad Prism 5.01 software. *p* < 0.05 was considered statistically significant. *: *p* < 0.05 compared to control.

**Figure 4 ijms-25-09024-f004:**
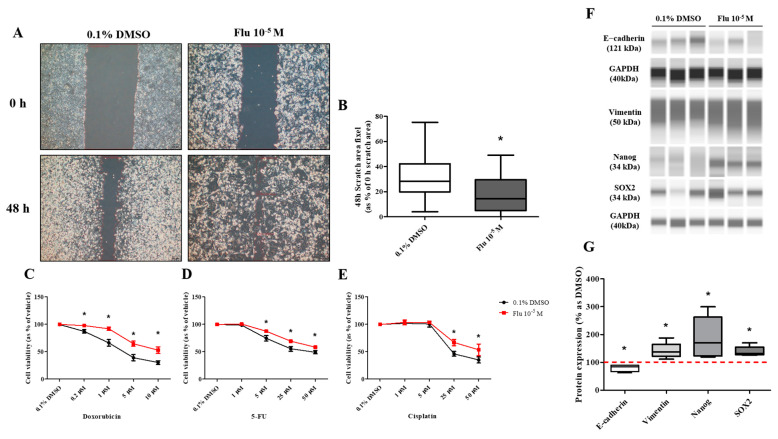
Effects of cell migration ability and chemo-resistance by applying the scratch assay and WST assay after treatment with 0.1% DMSO or fludioxonil in MDA-MB-231 breast cancer cells. (**A**) Cells were incubated for 2 days after seeding and scratched using a 1000 μL tip. After washing twice, the cells were treated by 0.1% DMSO or fludioxonil in the media for 48 h. (**B**) The value was presented in the graph. Cells were incubated for 2 days after seeding and treated each with (**C**) doxorubicin, (**D**) 5-FU, and (**E**) cisplatin for 72 h to confirm the chemo-resistance. (**F**,**G**) The changes of epithelial marker-related proteins were analyzed using Jess from Protein Simple. Flu: fludioxonil, DMSO was used as the vehicle control, and the value of the control containing 0.1% DMSO was set as 100%. Data in the graphs are obtained from at least three repeated experiments and are presented as the mean ± SEM. Data in the graphs are obtained from at least three repeated experiments and are presented as the mean ± SEM. Statistical analysis was determined by (**B**) one-way ANOVA followed by Dunnett’s multiple comparison, (**C**–**E**) two-way ANOVA followed by Bonferroni posttests, and (**G**) *t*-test (paired test) using the GraphPad Prism 5.01 software. *p* < 0.05 was considered statistically significant. *: *p* < 0.05 compared to control. Scale bar = 200 μm.

**Figure 5 ijms-25-09024-f005:**
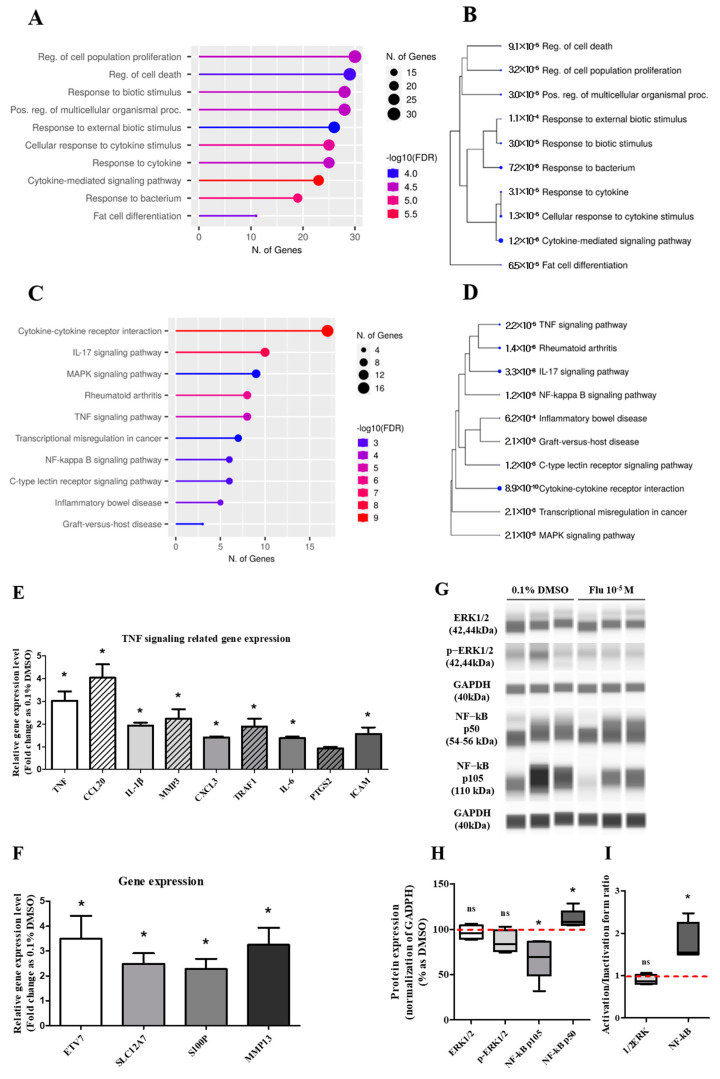
Gene Ontology enrichment analysis of DEGs and qPCR of related genes, and the expression change of TNF pathway-related proteins after treatment with 0.1% DMSO or fludioxonil in MDA-MB-231 breast cancer cells. The DEGs treated with 0.1% DMSO or fludioxonil were analyzed for enrichment in three ontologies: (**A**,**B**) GO biological process and (**C**,**D**) KEGG pathway using Shiny GO 0.77. (**E**) The change in TNF signaling-related gene expression was confirmed using real-time PCR. (**F**) Based on the results of DEGs, the cancer-related gene with a high expression of fold change was confirmed using real-time PCR. (**G**,**H**) The change in major pathway-related proteins expression was confirmed using Jess from Protein Simple. (**I**) The activity of 1/2ERK and NF-κB was separately graphed as the ratio of inactivation form to active form. Flu: fludioxonil, DMSO was used as the vehicle control, and the value of the control containing 0.1% DMSO was set as 100%. Data in the graphs are obtained from at least three repeated experiments and are presented as the mean ± SEM. Statistical analysis was determined by (**E**,**F**) *t*-test (paired test) using the GraphPad Prism 5.01 software. *p* < 0.05 was considered statistically significant. *: *p* < 0.05 compared to control.

**Figure 6 ijms-25-09024-f006:**
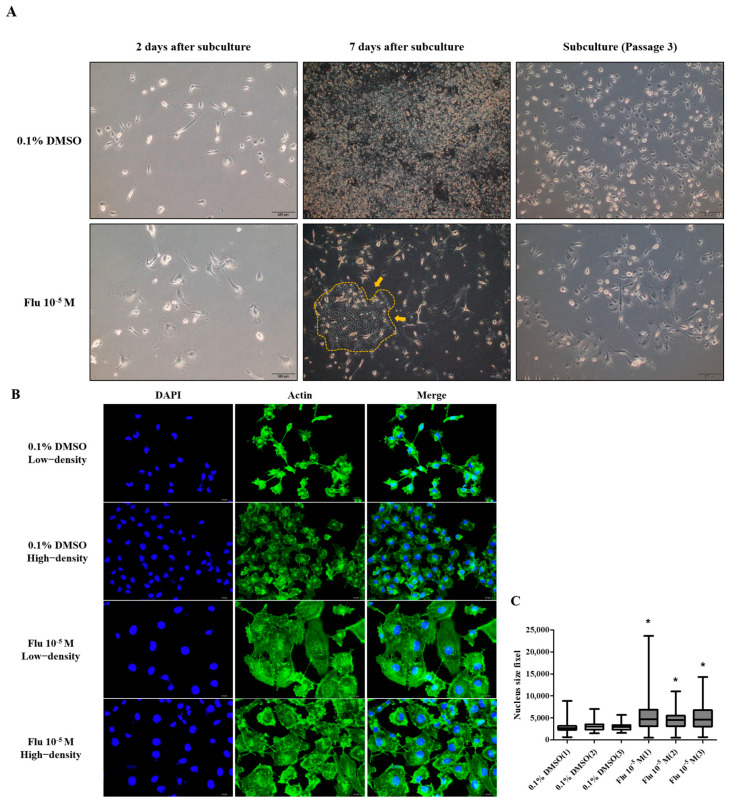
The acquisition process and morphology changes of daughter cells after treatment with 0.1% DMSO or fludioxonil in MDA-MB-231 breast cancer cells. (**A**) The daughter cells were passage-3 cells after being exposed to 0.1% DMSO or fludioxonil for 72 h. The cells were subcultured 1.0 × 10^5^ cells/dish and incubated for 7 days. This process was repeated twice. The yellow line is borderline of small colony that newly formed after treatment of fludioxonil for 72h. (**B**) The images of daughter cells were of morphology following different cell densities. (**C**) The graph was present the difference of nuclear size after subculturing three times following exposure of 0.1% DMSO or fludioxonil. Flu: fludioxonil, DMSO was used as the vehicle control, and the value of the control containing 0.1% DMSO was set as 100%. Data in the graphs are obtained from at least three repeated experiments and are presented as the mean ± SEM. Statistical analysis was determined by (**C**) one-way ANOVA followed by Dunnett’s multiple comparison using the GraphPad Prism 5.01 software. *p* < 0.05 was considered statistically significant. *: *p* < 0.05 compared to control. Upper image scale bar = 200 μm, lower image scale bar = 50 μm.

**Figure 7 ijms-25-09024-f007:**
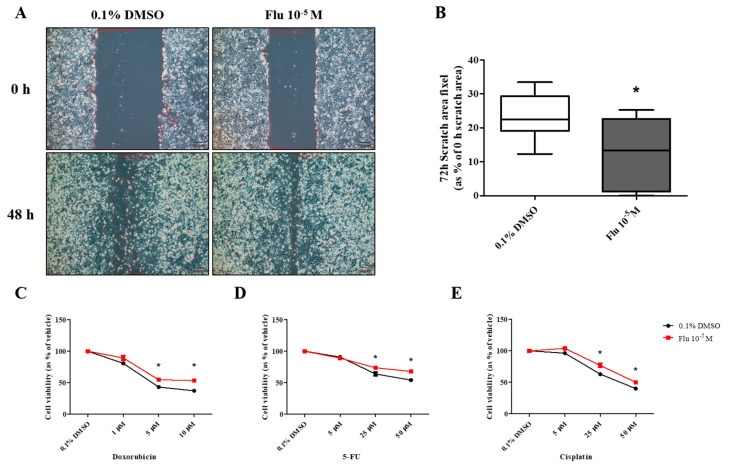
Effects of cell migration ability and chemo-resistance by applying the scratch assay and WST assay on daughter cells of MDA-MB-231 breast cancer cells after treatment with 0.1% DMSO or fludioxonil for 72 h. (**A**) Cells were incubated for 2 days after seeding and scratched using a 1000 μL tip. After washing twice, the cells were incubated for 48 h, (**B**) and the value was presented in the graph. To establish the chemo-resistance, cells were incubated for 2 days after seeding and treated each with (**C**) doxorubicin, (**D**) 5-FU, and (**E**) cisplatin for 72 h. Flu: fludioxonil, DMSO was used as the vehicle control, and the value of the control containing 0.1% DMSO was set as 100%. Data in the graphs are obtained from at least three repeated experiments and are presented as the mean ± SEM. Statistical analysis was determined by (**B**) one-way ANOVA followed by Dunnett’s multiple comparison and (**C**–**E**) two-way ANOVA followed by Bonferroni posttests using the GraphPad Prism 5.01 software. *p* < 0.05 was considered statistically significant. *: *p* < 0.05 compared to control. Scale bar = 50 μm or 100 pixels.

**Figure 8 ijms-25-09024-f008:**
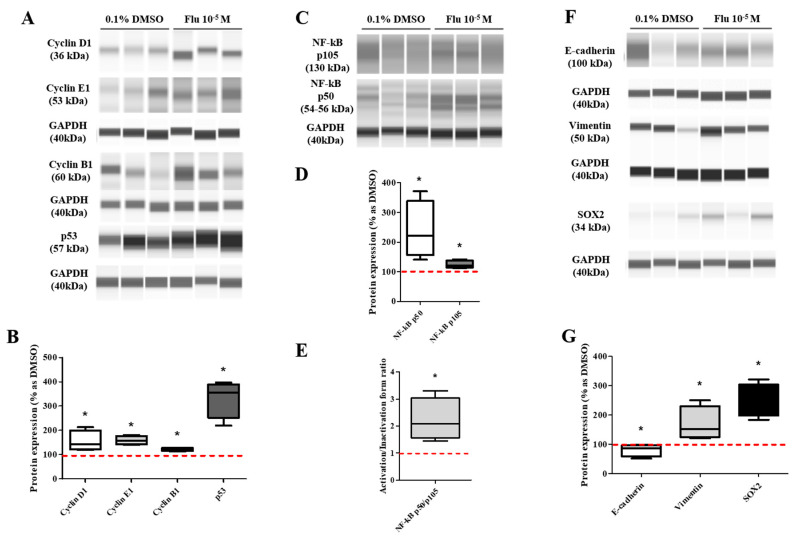
Effects of protein expression on daughter cells of MDA-MB-231 breast cancer cells. These cells were analyzed for protein expressions such as cell cycle-related (**A**,**B**), NF-κB-related (**C**–**E**), EMT-related and stemness-related proteins (**F**,**G**). Flu: fludioxonil, DMSO was used as the vehicle control, and the value of the control containing 0.1% DMSO was set as 100%. Data in the graphs are obtained from at least three repeated experiments and are presented as the mean ± SEM. Statistical analysis was determined by (**B**–**G**) *t*-test (paired test) compare with 0.1% DMSO using the GraphPad Prism 5.01 software. *p* < 0.05 was considered statistically significant. *: *p* < 0.05 compared to control.

**Figure 9 ijms-25-09024-f009:**
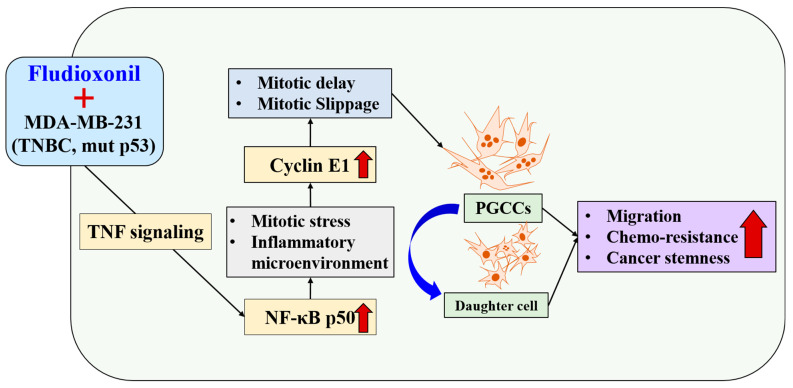
Fludioxonil induced the transformation process to PGCCs on p53mut MDA-MB-231 breast cancer cells via TNF with NF-κB signaling. Fludioxonil caused accumulation of the Cyclin E1 and promotion of the inflammatory cytokine-enriched microenvironment via up-regulation of TNF and NF-κB, and it led to the transformation to PGCCs via mitotic delay and mitotic slippage in mut p53mut TNBC MDA-MB-231 cells. PGCCs and their daughter cells had outstanding migration ability, chemo-resistance, and cancer stemness (red arrow: increasing of expression or ability, blue arrow: production of daughter cell in PGCCs).

**Table 2 ijms-25-09024-t002:** Composition of cell culture media.

Media Composition	Cell Culture Media Cell Seeding Media	Phenol-Free Media
Medium	DMEM	DMEM
Phenol red	Present	Absent
Glucose concentration	High (4500 mg/L)	Low (1000 mg/L)
FBS	5%	10% CD-FBS
Penicillin/streptomycin	2%	2%
HEPES	2% (20 mM)	2% (20 mM)
L-glutamine	4 mM	1 mM

**Table 3 ijms-25-09024-t003:** Schedules of cell culture in all experiments.

Day	1	2	3	5	6	7	9	10
WST assay Staining Cell cycle analysis RNA extraction Protein extraction	Seeding (plate)	MC (p/f)	CT	MC	Exp	-	-	-
Live-cell image	Seeding (plate)	MC (p/f)	CT and live start	MC	End	-	-	-
Migration assay	Seeding (90 mm)	MC (p/f)	CT	MC	Tryp&S (plate)	MMC and scratch	Exp	-
Anticancer drug resistance	Seeding (90 mm)	MC (p/f)	CT	MC	Tryp&S (plate)	ACD	-	Exp

MC(p/f): media change (phenol-free media) excluding chemicals, MC: media change (phenol-free media) including chemicals, CT: chemical treatment, Tryp&S: trypsinization and seeding, MMC: mitomycin C, Exp: experiment. ACD: treatment of anticancer drug.

**Table 4 ijms-25-09024-t004:** List of primers in real-time PCR.

Genes	Primer Bank ID		Sequence (5′-3′)	Tm (°C)	Size (bp)
TNF	25952110c1	F	CCTCTCTCTAATCAGCCCTCTG	60.8	220
		R	GAGGACCTGGGAGTAGATGAG	60.2	
CCL20	4759076a1	F	TGCTGTACCAAGAGTTTGCTC	60.2	220
		R	CGCACACAGACAACTTTTTCTTT	60.4	
IL-1β	27894305c1	F	ATGATGGCTTATTACAGTGGCAA	60.0	132
		R	GTCGGAGATTCGTAGCTGGA	60.8	
MMP3	73808272c1	F	AGTCTTCCAATCCTACTGTTGCT	61.0	226
		R	TCCCCGTCACCTCCAATCC	63.0	
CXCL3	4504157a1	F	CGCCCAAACCGAAGTCATAG	60.8	109
		R	GCTCCCCTTGTTCAGTATCTTTT	60.2	
TRAF1	300193044c1	F	TCCTGTGGAAGATCACCAATGT	61.0	117
		R	GCAGGCACAACTTGTAGCC	61.3	
IL-6	224831235c1	F	ACTCACCTCTTCAGAACGAATTG	60.2	149
		R	CCATCTTTGGAAGGTTCAGGTTG	61.3	
PTGS2	223941909c1	F	CTGGCGCTCAGCCATACAG	62.8	94
		R	CGCACTTATACTGGTCAAATCCC	61.0	
ICAM	167466197c1	F	ATGCCCAGACATCTGTGTCC	61.6	112
		R	GGGGTCTCTATGCCCAACAA	61.3	
ETV7	333470740c1	F	CTGCTGTGGGATTACGTGTATC	60.5	138
		R	GTTCTTGTGATTTCCCCAGAGTC	60.8	
SLC12A7	123701899c1	F	CTGGCGGGTCCTACTACATGA	62.8	127
		R	AAAATCTCGATGGTCCCCAAAAT	60.2	
S100P	45827727c1	F	AAGGATGCCGTGGATAAATTGC	61.3	79
		R	ACACGATGAACTCACTGAAGTC	60.0	
MMP13	296010793c1	F	ACTGAGAGGCTCCGAGAAATG	61.3	103
		R	GAACCCCGCATCTTGGCTT	62.7	
18s-rRNA	14165467c1	F	GCGGCGGAAAATAGCCTTTG	62.3	139
		R	GATCACACGTTCCACCTCATC	60.4	

**Table 5 ijms-25-09024-t005:** List of antibodies used in Western blot assay.

Protein	Manufacturer	Protein Size	Cat. No.	Dilution Ratio
Cyclin D1	Biolegend	36	681902	1:20
Cyclin E1	Santacruz	53	sc-377100	1:20
Cyclin A1	Bioss	51	bs-5739R	1:10
Cyclin B1	Biolegend	48	647901	1:20
p53	Santacruz	53	sc-126	1:10
E-cadherin	Biolegend	100	324102	1:20
Vimentin	Thermo Fisher	58	MA5-11883	1:20
Nanog	Bioss	34	bs-0829R	1:10
SOX2	Biolegend	35	656102	1:20
NF-κB p50(p105)	Biolegend	50,100	616701	1:20
ERK1/2	Cell signaling	42,44	4695T	1:10
p-ERK1/2 (Thr202/Tyr204)	Biolegend	42,44	369501	1:10
GAPDH	Abcam	40	ab8245	1:100

Biolegend (San Diego, CA, USA), Santacruz (Santa Cruz, CA, USA), Bioss (Woburn, MC, USA), Cell signaling (Cell signaling Technology, Danvers, MA, USA), Abcam (Waltham, MA, USA).

## Data Availability

Data are contained within the article and Appendix A.

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
