# Peer review of "A Fungicide, Fludioxonil, Formed the Polyploid Giant Cancer Cells and Induced Metastasis and Stemness in MDA-MB-231 Triple-Negative Breast Cancer Cells"

_ijms, 2024, doi:10.3390/ijms25169024_

Round 1

Reviewer 1 Report

Comments and Suggestions for Authors

The bioavailability and release of Fludioxonil  should be done.

- How to approve  the formation of PGCCs in tumors of fludioxonil exposed mice.

-The injection of PGCCs to form xenografted mouse model is not clear

- High Fludioxonil concentration compared with actually exposed 

concentration on human.

What is the fate of fludioxonil and metabolism and role of CYP450.

Comments on the Quality of English Language

The bioavailability and release of Fludioxonil  should be done.

- How to approve  the formation of PGCCs in tumors of fludioxonil exposed mice.

-The injection of PGCCs to form xenografted mouse model is not clear

- High Fludioxonil concentration compared with actually exposed 

concentration on human.

What is the fate of fludioxonil and metabolism and role of CYP450.

Author Response

Reviewer 1

The bioavailability and release of Fludioxonil should be done.

à Following the reviewer’s comments, the bioavailability and release of Fludioxonil is important information in vivo. Fludioxonil has already been reported for residue concentration and toxicology by the Joint FAO/WHO meeting on Pesticide Residues (JMPR). 

 The half-life of Fludioxonil is approximately 2-4 hours in the atmosphere by hydroxyl radical oxidation reported by JMPR in 2004. In the animal metabolism test using the goats, Fludioxonil was administered orally at 150 mg/day and excreted in the feces (50 to 60%) and in the urine (15 to 23%). The total recovery of Fludioxonil by excretion was 94 to 98%. However, the highest residues remained in the liver and kidneys, especially residues of Fludioxonil in the fat were 2-4 times in the muscle, and residues in the milk were 1.8 times on day 4. In addition, the residue of Fludioxonil was kept high in the plant (JMPR, 2005). According to JMPR in 2022, the findings of the newly submitted pharmacokinetic study of Fludioxonil in rats are consistent with the conclusions of the 2004 JMPR evaluation (Pesticide residues in food 2022, JMPR).

 Although fruits and vegetables washed with the tap water have reduced levels of residual pesticides, high-risk groups such as small children, residents of rural areas, and farm workers may face health hazards associated with the accumulation of residue pesticides.

 Therefore, although the outcomes of the present study indicate that Fludioxonil can induce breast cancer progression by promoting the PGCCs process and metastatic behaviors, more considerate settings for dose levels of the compound reflecting human and environmental exposure context will be needed in further studies.

- How to approve the formation of PGCCs in tumors of Fludioxonil exposed mice. The injection of PGCCs to form xenografted mouse model is not clear.

à In this study, the authors did not present in vivo results to confirm the effect of Fludioxonil in a mouse model. But, like a reviewer’s comments, the authors had undergone trial and error to set up the mouse models for evaluation of PGCCs in breast tumor.

First. The authors conducted in vivo experiments, in which the PGCCs formed by Fludioxonil exposure were injected in mics. As per the methodology, the mouse was injected with the mixture of Matrigel and PGCCs exposed to Fludioxonil. The tumor of PGCC injected xenografted mice were grown rapidly and larger than the control. These were softer than the tumors of the control group and were puss filled. The authors concluded that the rapid growth of PGCCs had led to necrosis in the tumor. Therefore, the authors needed further studies for objective histological comparison with the control group.

Second. The authors made an in vivo plan to test the spontaneous production of PGCC in xenografted mouse models. But, because the range of Fludioxonil concentration and exposure period is too uncertain, significant mouse sacrifice was predicted. For this reason, the authors didn't conduct the in vivo test.

Further studies of Fludioxonil are underway, and authors are gathering further evidence. If found necessary, the authors will develop an in vivo model as suggested by the reviewer for our ongoing study.

- High Fludioxonil concentration compared with actually exposed concentration on human.

à The authors believe that the purpose of the Fludioxonil biological concentration in nature and the concentration used in this paper are different. The concentration used in this paper is high concentrations to identify potential effects on the body, and no cohort study has yet been conducted to confirm the effects on actual organisms. In addition, many factors must be considered to compare the concentration of Fludioxonil in an actual body and cells. If the factors are not completely considered, an error may be derived.

Studies of BPA (Vincent Le Fol et al (2017), In vitro and in vivo estrogenic activity of BPA, BPF, and BPS in zebrafish-specific assays. Ecotoxicology and Environmental Safety, 142, 150-156) known as EDCs for representative ER, are conducted in several studies at 10-5M showing biologically significant changes, not concentrations exposed to actual users. This study confirmed the formation of PGCC at 10-5M, which indicates that Fludioxonil affects the development of breast cancer cell lines.

What is the fate of Fludioxonil and metabolism and role of CYP450.

à In the 2022 JMPR report, the chemical structures of Fludioxonil metabolites are presented, and each toxicity is compared with Fludioxonil. Almost all metabolites are found in plants and have not detected toxicity. But partial metabolites are found in animal bodies and have toxicity similar to Fludioxonil, which induces chromosomal aberrations in vitro (Bohnenberger S et al., 2007). In a study to confirm the in vitro metabolic profiling of Fludioxonil, each liver microsome and a NADPH-regenerating system in male and female human and Wistar rats were exposed to Fludioxonil. As a result, the minor metabolites were partially detected in liver microsomes of humans and rats, but there were no human-specific metabolites. The individual microsomal cytochrome P450 enzymes(CYP450s) responsible for the metabolism of Fludioxonil were not identified (Thibaut R et al, 2017).

AhR, as an aryl-hydrocarbon receptor, is the regulator of enzymes such as cytochrome P450s). Another study investigated that Fludioxonil is AhR-agonist (Medjakovic S et al.  Environ Toxicol. 2014;29(10):1201-1216). In addition, in another study, an inhibition of the metabolic enzyme CYP3A4 by Fludioxonil was identified in liver cells (Lasch A, et al. Arch Toxicol. 2021;95(4):1397-1411).

These studies are everything about the Fludioxonil in the liver and indicate that there is currently a lack of research on the metabolism and action of Fludioxonil in the liver. The authors' study will provide evidence for the negative effects of Fludioxonil onthe live body and stimulate the curiosity of scientists to conduct research on Fludioxonil.

Reviewer 2 Report

Comments and Suggestions for Authors

The manuscript deals with very interesting and important topic; use of pesticides effects every aspect of human health and therefore, it is of great importance to elucidate every possible mechanism of its effect.

The authors did a great job; the research is well designed, materials and methods are described very detailed, results are very well presented. Every question raised while reading a manuscript was answered in next paragraph. Finally, discussion observes all results obtained in this research and similar researcher and limitations of the study are clearly listed.

After all, I don’t have severe concerns about the manuscript. In contrast, I’m impressed with overall research, number of methods used, presentation of the results and the flow of the manuscript.

My only comment is that the most of the figures are not of good quality and text on them is hardly readable.

Furthermore, I would like authors to comment the result of increase of cells viability after treatment with Flu in T47D and MCF-7 cells comparing to MDA-MB-231 (triple negative ones); could the reason may be the mut-p53 or wild type-p53 in these cell lines, or simply presence of estrogen receptor?

Thank you

Author Response

Reviewer 2

The manuscript deals with very interesting and important topic; use of pesticides effects every aspect of human health and therefore, it is of great importance to elucidate every possible mechanism of its effect.

The authors did a great job; the research is well designed, materials and methods are described very detailed, results are very well presented. Every question raised while reading a manuscript was answered in next paragraph. Finally, discussion observes all results obtained in this research and similar researcher and limitations of the study are clearly listed.

After all, I don’t have severe concerns about the manuscript. In contrast, I’m impressed with overall research, number of methods used, presentation of the results and the flow of the manuscript.

à We have really appreciated the reviewer’s comments to complete this study.

My only comment is that the most of the figures are not of good quality and text on them is hardly readable.

à First of all, the authors are sorry for the reviewer’s inconvenience. We used the 'PowerPoint Program' to send the figures in high resolution to the Intl J Mol Sci Editor. Based on experience, the figures in the 'PDF file' sent to the reviewers were probably low resolution.

Following the reviewers' comments, the authors will check high-resolution images and text in figures with the editorial team of Intl J Mol Sci.

Furthermore, I would like authors to comment the result of increase of cells viability after treatment with Flu in T47D and MCF-7 cells comparing to MDA-MB-231 (triple negative ones); could the reason may be the mut-p53 or wild type-p53 in these cell lines, or simply presence of estrogen receptor?

à In the previous author's study, Fludioxonil is revealed to have an estrogen effect and promotes the growth of breast cancer cells. Especially the effect of Fludioxonil is confirmed to be higher in vivo (xenografted mouse) than in vitro (Go RE, et al. Environ Toxicol. 2017;32(4):1439-1454). In addition, T47D breast cancer is much-p53 type and has the estrogen receptor, but cell viability is promoted by Fludioxonil in this study. In addition, the results of T47D breast cancer are similar to MCF-7 breast cancer, which has estrogen receptors and willd-type p53 genes.

Although further studies are needed, Fludioxonil probably has a diverse effect on our bodies. Intensive research is needed on the diverse causes, including the relationships between estrogen receptors, p53 gene types, and Fludioxonil. However, this study suggests that PGCCs are induced by mut-p53 in the absence of estrogen receptors.

Round 2

Reviewer 1 Report

Comments and Suggestions for Authors

Extenisve English editing 

Comments on the Quality of English Language

Extenisve English editing 

Author Response

Reviewer Comments

Reviewer 1

The bioavailability and release of Fludioxonil should be done.

à Following the reviewer’s comments, the bioavailability and release of Fludioxonil is important information in vivo. Fludioxonil has already been reported for residue concentration and toxicology by the Joint FAO/WHO meeting on Pesticide Residues (JMPR). 

 The half-life of Fludioxonil is approximately 2-4 hours in the atmosphere by hydroxyl radical oxidation reported by JMPR in 2004. In the animal metabolism test using the goats, Fludioxonil was administered orally at 150 mg/day and excreted in the feces (50 to 60%) and in the urine (15 to 23%). The total recovery of Fludioxonil by excretion was 94 to 98%. However, the highest residues remained in the liver and kidneys, especially residues of Fludioxonil in the fat were 2-4 times in the muscle, and residues in the milk were 1.8 times on day 4. In addition, the residue of Fludioxonil was kept high in the plant (JMPR, 2005). According to JMPR in 2022, the findings of the newly submitted pharmacokinetic study of Fludioxonil in rats are consistent with the conclusions of the 2004 JMPR evaluation (Pesticide residues in food 2022, JMPR).

 Although fruits and vegetables washed with the tap water have reduced levels of residual pesticides, high-risk groups such as small children, residents of rural areas, and farm workers may face health hazards associated with the accumulation of residue pesticides.

 Therefore, although the outcomes of the present study indicate that Fludioxonil can induce breast cancer progression by promoting the PGCCs process and metastatic behaviors, more considerate settings for dose levels of the compound reflecting human and environmental exposure context will be needed in further studies.

- How to approve the formation of PGCCs in tumors of Fludioxonil exposed mice. The injection of PGCCs to form xenografted mouse model is not clear.

à In this study, the authors did not present in vivo results to confirm the effect of Fludioxonil in a mouse model. But, like a reviewer’s comments, the authors had undergone trial and error to set up the mouse models for evaluation of PGCCs in breast tumor.

First. The authors conducted in vivo experiments, in which the PGCCs formed by Fludioxonil exposure were injected in mics. As per the methodology, the mouse was injected with the mixture of Matrigel and PGCCs exposed to Fludioxonil. The tumor of PGCC injected xenografted mice were grown rapidly and larger than the control. These were softer than the tumors of the control group and were puss filled. The authors concluded that the rapid growth of PGCCs had led to necrosis in the tumor. Therefore, the authors needed further studies for objective histological comparison with the control group.

Second. The authors made an in vivo plan to test the spontaneous production of PGCC in xenografted mouse models. But, because the range of Fludioxonil concentration and exposure period is too uncertain, significant mouse sacrifice was predicted. For this reason, the authors didn't conduct the in vivo test.

Further studies of Fludioxonil are underway, and authors are gathering further evidence. If found necessary, the authors will develop an in vivo model as suggested by the reviewer for our ongoing study.

- High Fludioxonil concentration compared with actually exposed concentration on human.

à The authors believe that the purpose of the Fludioxonil biological concentration in nature and the concentration used in this paper are different. The concentration used in this paper is high concentrations to identify potential effects on the body, and no cohort study has yet been conducted to confirm the effects on actual organisms. In addition, many factors must be considered to compare the concentration of Fludioxonil in an actual body and cells. If the factors are not completely considered, an error may be derived.

Studies of BPA (Vincent Le Fol et al (2017), In vitro and in vivo estrogenic activity of BPA, BPF, and BPS in zebrafish-specific assays. Ecotoxicology and Environmental Safety, 142, 150-156) known as EDCs for representative ER, are conducted in several studies at 10-5M showing biologically significant changes, not concentrations exposed to actual users. This study confirmed the formation of PGCC at 10-5M, which indicates that Fludioxonil affects the development of breast cancer cell lines.

What is the fate of Fludioxonil and metabolism and role of CYP450.

à In the 2022 JMPR report, the chemical structures of Fludioxonil metabolites are presented, and each toxicity is compared with Fludioxonil. Almost all metabolites are found in plants and have not detected toxicity. But partial metabolites are found in animal bodies and have toxicity similar to Fludioxonil, which induces chromosomal aberrations in vitro (Bohnenberger S et al., 2007). In a study to confirm the in vitro metabolic profiling of Fludioxonil, each liver microsome and a NADPH-regenerating system in male and female human and Wistar rats were exposed to Fludioxonil. As a result, the minor metabolites were partially detected in liver microsomes of humans and rats, but there were no human-specific metabolites. The individual microsomal cytochrome P450 enzymes(CYP450s) responsible for the metabolism of Fludioxonil were not identified (Thibaut R et al, 2017).

AhR, as an aryl-hydrocarbon receptor, is the regulator of enzymes such as cytochrome P450s. Another study investigated that Fludioxonil is AhR-agonist (Medjakovic S et al.  Environ Toxicol. 2014;29(10):1201-1216). In addition, in another study, an inhibition of the metabolic enzyme CYP3A4 by Fludioxonil was identified in liver cells (Lasch A, et al. Arch Toxicol. 2021;95(4):1397-1411).

These studies are everything about the Fludioxonil in the liver and indicate that there is currently a lack of research on the metabolism and action of Fludioxonil in the liver. The authors' study will provide evidence for the negative effects of Fludioxonil onthe live body and stimulate the curiosity of scientists to conduct research on Fludioxonil.